# Sex-Specific Effects on Total Body Fat Gain with 4-Week Daily Dosing of Raspberry Ketone [4-(4-Hydroxyphenyl)-2-butanone] and Ketogenic Diet in Mice

**DOI:** 10.3390/nu15071630

**Published:** 2023-03-28

**Authors:** Lihong Hao, Nicholas T. Bello

**Affiliations:** Department of Animal Sciences, School of Environmental and Biological Sciences, Rutgers, The State University of New Jersey, New Brunswick, NJ 08901, USA

**Keywords:** ketogenesis, [4-(4-hydroxyphenyl)-2-butanone], microbiome, glucose intolerance, sex effect

## Abstract

**Background**: Raspberry ketone (RK: [4-(4-Hydroxyphenyl)-2-butanone]) is a dietary supplement marketed for weight control. RK is structurally unrelated to the ketone bodies elevated with a ketogenic diet (KD). This study aims to determine whether RK oral supplementation with KD improves the weight loss outcomes in high-fat diet (HFD; 45% fat)-fed mice. **Methods**: Male and female C57BL/6J mice were HFD-fed for 9 weeks and switched to KD (80% fat) or a control diet (CD; 10% fat) or continued with the HFD for 4 weeks. Coincident with the diet switch, each diet group received oral RK (200 mg/kg/day) or a vehicle. **Results**: In male KD-fed mice, oral RK reduced body weight by ~6% (KD_Veh: −9.2 ± 1% vs. KD_RK: −15.1 ± 1%) and fat composition by ~18% (KD_Veh: −16.0 ± 4% vs. KD_RK: −34.2 ± 5%). HFD and KD feeding induced glucose intolerance in both male and female mice. Oral RK decreased the glucose area under the curve in female mice by ~6% (KD_Veh: 44,877 ± 957 vs. KD_RK: 42,040 ± 675 mg*min/dl). KD also had gut microbiota alterations with higher alpha diversity in males and more beta diversity with RK. These findings suggest sex-specific weight loss effects with RK and KD in mice.

## 1. Introduction

Raspberry ketone (RK; [4-(4-hydroxyphenyl)-2-butanone]) is a naturally-occurring aromatic phenolic compound associated with the flavor and aroma of red raspberries [1]. RK is expensive to produce from natural sources, but inexpensively synthesized by a two-step reaction from acetone, phenol, and formaldehyde [2]. In the US, RK is a flavoring agent permitted by the FDA for direct addition to food for human consumption [3,4]. In recent years, RK has been marketed as a dietary supplement for weight loss [4]. However, rodent studies have shown that RK prevents body weight gain, suppresses fat accumulation, and increases fat oxidation and lipolysis, suggesting a potential to maintain weight loss [5,6,7,8,9,10]. Furthermore, in vitro and in silico enzyme kinetics studies suggest that RK may act as an α-glucosidase inhibitor to inhibit carbohydrate hydrolysis and potentially reduce postprandial blood glucose levels [11]. In vivo studies in normal weight and overweight mice suggest that RK normalizes the glucose tolerance impairment by high-fat diet feeding [5,7]. Nonetheless, there are no human studies to indicate that RK as a single ingredient dietary supplement improves metabolic outcomes associated with obesity or promotes weight loss [12,13].

A ketogenic diet (KD) is a low-carbohydrate and high-fat diet. It promotes the production of ketone bodies and fatty acid oxidation and has been demonstrated as a dietary intervention in weight loss in humans [14,15,16]. A KD improves glucose homeostasis, insulin sensitivity and lipid profiles in obese and type 2 diabetic patients [14,16,17]. Rodent studies have also suggested the improvement of glucose tolerance in genetically or diet-induced obese mice [18]. However, other studies have suggested that a KD induces glucose intolerance, reduces beta cell mass, and causes insulin resistance [19,20,21,22]. The effects of a KD on weight loss in obesity are still not fully characterized and could be related to changes in the gut microbiota [23].

Based on the popularity of the ketogenic diet for weight loss, RK has been marketed as a natural ketone. RK, however, has no known interactions with the production or enhancement of ketone bodies (i.e., β-hydroxybutyrate, acetoacetate, and acetone).

In this study, we sought to investigate whether dietary supplementation of RK improves KD outcomes on body weight loss and glucose intolerance associated with high-fat diet weight gain. Aside from investigating the notion that RK has a “ketone” effect on KD outcomes, one possibility is that RK supplementation could have an additive effect on KD weight loss and glucose regulation outcomes. This study is the first study to investigate whether RK improves the weight loss effects of KD in male and female mice maintained on a high-fat diet (HFD). To mimic how dietary supplements are consumed, RK was administered by oral gavage at a dose (200 mg/kg) previously demonstrated to prevent body weight gain on an HFD [5,6,7]. In addition, because the effects of KD on gut microbiota could contribute to the weight loss potential of KD, we also examined the interaction of RK and KD on microbiota diversity and structure.

## 2. Materials and Methods

### 2.1. Mice

A total of 109 male and female C57BL/6J mice (7-week-old) were purchased from The Jackson Laboratory (Cat#000664; Bar Harbor, ME, USA). All the mice were housed, four mice per cage, upon arrival and were acclimated on standard chow (Purina Mouse Diet 5015, 25.34% fat, 19.81% protein, 54.86% CHO, 3.7 kcal/g; Lab Diet, St. Louis, MO, USA) for 10 days. All mice had free access to water, unless otherwise noted, and maintained on a 12-h light and 12-h dark cycle with lights on from 500 to 1700 h. The animal care protocol was approved by the Institutional Animal Care and Use Committee of Rutgers University (OLAW #A3262-01); Protocol #999900014.

All mice were fed a high-fat diet (HFD; 45%fat) for 9 weeks after acclimation and subjected to a daily dosing and diet regimen for 4 weeks (see Section 2.2). Blood ketone measurements, oral glucose tolerance test, body composition assessment, fecal collection and microbiota analysis were performed during the 4-week daily dosing and diet regimen. Blood and tissue samples were collected for plasma biomarkers and gene expression analysis (Figure 1).

### 2.2. Daily Dosing and Diet Regimen

Thirty-nine (39) male and thirty-eight (38) female mice were fed a high-fat diet (HFD; D12451, Research Diets, Inc., New Brunswick, NJ, USA; 45% fat, 20% protein, 35% carbohydrate; 4.73 kcal/g) for 9 weeks. Food intake, body weight and water intake were measured weekly. Body composition was measured at the end of week 9. All male mice or female mice were equally divided by body weight and fat mass into four groups of 9 to 10 mice per each group. After 9 weeks of feeding the mice an HFD, half of the mice of each sex were switched to a ketogenic diet (KD; D03022101, Research Diets, Inc., New Brunswick, NJ, USA; 80% fat, 20% protein, 0.1% carbohydrate; 6.14 kcal/g) and another half of the mice in each sex were switched to a low-fat control diet (CD; D12450K; 10% fat, 20% protein, 70% carbohydrate; 3.85 kcal/g) for 4 weeks. Coincident with the diet switch, daily oral dosing was initiated in mice with the following treatments: vehicle (Veh; 50% propylene glycol, 40% water, and 10% dimethyl sulfoxide; DMSO) or raspberry ketone (RK, 200 mg/kg; 4[4-hydroxyphenyl]-2-butanone; 99%; cat#178519; Sigma Aldrich, Inc., St. Louis, MO, USA).

In another separate group of male (n = 16) and female (n = 16) mice, following a 9-week HFD feeding, all mice in each sex were equally divided by body weight and fat mass into either Veh or RK group (8 mice/group/sex) and were maintained on HFD for 4 weeks. Oral gavage was performed using single-use, sterile plastic feeding tubes (20 ga × 30 mm; cat# FTP-20-30, Instech Laboratories, Plymouth Meeting, PA, USA).

### 2.3. Blood Ketone Measurement

After 5 h of fasting, blood ketone (β-hydroxybutyrate) levels were measured in blood samples collected from the tip of the tail vein using a blood ketone meter (Precision Xtra; Abbott Diabetes Care Inc., Alameda, CA, USA). Blood ketone was measured after 8 weeks of HFD feeding (baseline) and 3 weeks of oral gavage and diet regimen.

### 2.4. Oral Glucose Tolerance Test

The mice were fasted in new clean cages for 5 h. Blood samples were collected from the tip of the tail vein, and baseline blood glucose levels were measured using a blood glucose meter (Clarity BG1000; Clarity Diagnostics, LLC, Boca Raton, FL, USA). Mice were then given a dose of glucose (2 g/kg) orally, and blood glucose levels were measured at 15, 30, 60, 90, 120, and 180 min. All mice were returned to their home cages and given food and water immediately after the oral glucose tolerance test (OGTT), but mice were not administered their respective daily treatments of RK or Veh. Oral dosing of compounds was resumed the next day after the OGTT. An oral glucose tolerance test was conducted after 3 weeks of oral gavage and diet regimen.

### 2.5. Body Composition

The mice’s lean, fat, and fluid mass were measured using an EchoMRI 3-in-1 (Echo Medical Systems, Houston, TX, USA) body composition analyzer. Body composition was assessed after 8 weeks of HFD feeding (baseline) and 4 weeks of oral gavage and diet regimen.

### 2.6. Fecal Collection and Microbiota Analysis

Fresh fecal pellets were collected after 3 weeks of oral dosing and diet regimen from a subgroup (n = 40) of RK or vehicle-dosed KD and CD mice (n = 5/group, n = 20/sex), snap-frozen in liquid nitrogen and stored at −80 °C until analysis. The gut microbiota analysis was performed at Rutgers Center for Microbiome Analysis. Briefly, genomic DNA was extracted [24], and hypervariable region V4 of the 16S rRNA was amplified using fusion primers (i.e., 515F and 806R V4 primer pair) [25,26] ligated to adapters and barcodes from ThermoFisher Scientific in polymerase chain reaction (PCR), amplicons were quantified and sequenced using the Ion GeneStudio S5 (ThermoFisher Scientific, Waltham, MA, USA), microbiota alpha and beta diversity were analyzed. Overall gut microbiota structure was evaluated using the alpha diversity index (Shannon index) and beta diversity distance metric (Bray-Curtis distance). Principal coordinates analysis (PCoA) was performed to visualize differences in gut microbiota structure.

### 2.7. Blood and Tissue Collection and Analysis

All the mice were fasted in new clean cages for 5 h. The mice were then deeply anesthetized under isoflurane mixed with oxygen and exsanguinated, followed by decapitation. Blood was collected in EDTA tubes by cardiac puncture under deep isoflurane anesthesia. Proteinase inhibitors were added, and the plasma was prepared at 4 °C for plasma biomarkers analysis. Tissues were collected in RNase later and stored at −80 °C.

### 2.8. Plasma Biomarker Assay

Plasma apelin, insulin and leptin were analyzed using a commercially available apelin enzyme-linked immunosorbent assay (ELISA) kit, a rat/mouse insulin ELISA kit, and a mouse leptin ELISA kit (Millipore Sigma, St. Louis, MO, USA).

### 2.9. RNA Extraction, and Reverse Transcription and Quantitative Real-Time Polymerase Chain Reaction

Liver RNA was extracted using a Machery-Nagel Nucleospin RNA extraction kit (MACHEREY-NAGEL Inc., Allentown, PA, USA). DNase I was used to minimize any genomic DNA contamination. RNA quantity was determined using a NanoDrop ND-2000 spectrophotometer (Thermo Fisher Scientific Inc., Waltham, MA, USA).

Complementary DNA synthesis and quantitative real-time polymerase chain reaction (qPCR) were performed according to our previous protocol [5]. All gene expression data were expressed as an n-fold difference relative to the HFD_Veh group.

Primers for the following genes were purchased from BioRad Laboratories: peroxisome proliferator-activated receptor alpha (*Pparα*), peroxisome proliferator-activated receptor-gamma coactivator (PGC)-1alpha (*PGC1α*), fibroblast growth factor 21 (*Fgf21*), fibroblast growth factor receptor (*Fgfr1*). Primers for housekeeping genes Gapdh and Hprt were synthesized by Life Technologies, Inc. (Life Technologies, Inc., Carlsbad, CA, USA), and primer sequences were as follows:


*Gapdh:*


Forward: TGACGTGCCGCCTGGAGAAA;

Reverse: AGTG TAGCCCAAGATGCCCTTCAG.


*Hprt:*


Forward: GCTTGCTGGTGAAAAGGACCTCTCGAAG;

Reverse: CCCTGAAGTACTCATTATAGTCAAGGGCAT.

### 2.10. Statistical Analyses

Variables from both male and female mice, including body weight, body composition, diet intake, water intake, glucose, ketone and gene expression, were analyzed using three-way analysis of variance (ANOVA) using sex, diet, and dosing as independent variables. When there was a sex difference, a sex-specific analysis in male or female mice was conducted. We ran a two-way analysis of variance (ANOVA) using diet and dosing as independent variables, followed by the Newman-Keuls post hoc test when justified. The statistical analyses and graphing were performed using GraphPad Prism version 9.5 (GraphPad Software Inc.; Boston, MA, USA). All the data were represented as mean ± standard error of the mean (SEM), and a p-value less than 0.05 was considered statistically significant. For gut microbiota analysis, alpha diversity using the Shannon index as a representative metric was analyzed by the Kruskal-Wallis test followed by Dunn’s post hoc when justified. Beta diversity based on Bray Curtis distance was analyzed using Permutational Multivariate Analysis of Variance (PERMANOVA). Gut microbiota statistical analyses were performed at Rutgers Center for Microbiome Analysis using R version 4.1.1.

## 3. Results

### 3.1. Addition of Raspberry Ketone to Ketogenic Diet Fed Mice Promoted Weight Loss and Body Composition Change

All mice were fed an HFD (45%fat) for 9 weeks and switched to a ketogenic diet (KD, 80% fat) or Control Diet (CD, 10%fat) or continued on HFD for 4 weeks. Coincident with the diet switch, mice were daily dosed with vehicle (Veh) or raspberry ketone (RK; 200 mg/kg). Body weight or body composition change was used for comparison (Figure 1 and Appendix A). Our results showed that there was a sex difference for body weight gain [F (1, 97) = 116.0, *p* < 0.0001] and body composition measures, including fat [F (1,97) = 21.4, *p* < 0.0001] and lean [F (1, 97) = 8.4, *p* < 0.01] mass. Male mice had greater loss of body weight, fat and lean mass. There is also a sex effect for accumulated diet [F (1,97) = 74.6, *p* < 0.0001] and water intake [F (1,97) = 56.2, *p* < 0.0001]. Male mice had a higher calorie intake and water intake than female mice.

In a separate analysis in male mice, RK dosing significantly reduced body weight (*p* < 0.05), fat mass (*p* < 0.01) and lean mass (*p* < 0.05) compared to vehicle in KD mice (Figure 1A,C,E and Appendix A). However, oral RK dosing did not reduce weight or fat or lean mass in HFD or CD mice. In addition, there is a diet effect on weight loss and fat reduction: CD mice had more weight and fat loss compared to HFD or KD mice; although there is no significant diet effect on weight/fat loss between HFD and KD mice, the addition of RK to KD diet reduced body weight and fat mass compared to HFD mice. The dietary effect on weight loss may be due to the difference in energy intake. CD mice had lower accumulated calorie intakes compared to HFD and KD mice (*p* < 0.05; Figure 1G).

In females, RK had no effect on body weight, body composition and diet intake (Figure 1B,D,F,H and Appendix A). There is no significant diet effect on body weight and composition as seen in male mice, except that the CD mice had more fat loss than the KD mice and more lean loss than the HFD mice. The CD mice had lower calorie intakes compared to the HFD or KD mice. There is no difference in diet intake between the HFD or KD mice.

### 3.2. Blood Ketone Levels Were Raised by Ketogenic Diet Feeding but Glucose Intolerance Was Induced by High-Fat Diet Feeding in Overweight Male and Female Mice

Fasting blood glucose and ketone levels were measured, and oral glucose tolerance tests (OGTTs) were conducted after 3 weeks of diet feeding and daily dosing. Our results showed that there were effects of sex on baseline glucose [F (1,98) = 10.2, *p* < 0.005], glucose area under the curve (AUC) [F (1,98) = 36.9, *p* < 0.0001] and blood ketone levels [F (1,98) = 72.2, *p* < 0.0001]. Male mice had higher baseline glucose and glucose AUC, while female mice had higher blood ketone levels.

In a separate analysis of male mice, baseline glucose and ketone levels were affected by diet types (Figure 2 and Appendix A). Post hoc analysis showed that HFD and KD mice had higher baseline glucose levels than CD mice (Appendix A). The blood ketone levels in KD mice were higher than in HFD or CD mice (Figure 2E). The addition of RK had no significant effect on blood ketone levels compared to vehicles in each diet group. The glucose AUC for glucose response to the tolerance test was also affected by diet (Figure 2C). Post hoc analysis showed that the KD mice had a higher AUC than HFD or CD mice, and CD mice had the lowest AUC (*p* < 0.0002; Figure 2C). The addition of RK to diets had no significant effect on glucose or ketone levels when compared to vehicles in each diet group (Figure 2 and Appendix A).

In females, RK-dosed HFD or KD mice had higher baseline glucose (*p* < 0.01; Appendix A) compared to RK-dosed CD mice. There is no difference in baseline glucose levels between HFD and KD mice. Blood ketone levels were increased by KD diets compared to HFD or CD diets (*p* < 0.01; Figure 2F). For the glucose AUC in female mice, KD mice had a higher AUC than HFD or CD mice; CD mice had the lowest AUC (*p* < 0.0001; Figure 2D). The daily dosing of RK in KD mice significantly decreased the glucose AUC compared with vehicles (*p* < 0.05; Figure 2D); however, RK addition to HFD or CD had no effect on the glucose AUC in each diet group.

### 3.3. Raspberry Ketone and Diets Feeding Affect Hepatic Gene Expression, Plasma Apelin, Insulin and Leptin Differentially in Male and Female Mice

Based on the observation that KD diet feeding reduced the body weight/composition of sexes differently, impaired glucose tolerance and increased ketone levels, we next examined several hepatic gene expressions (Appendix A) and circulating hormones levels (Figure 3) that related to ketone production, glucose regulation, and lipid metabolism.

There was a sex difference for gene expression of *Pparα* [F (1,48) = 44.64, *p* < 0.0001], *PGC1α* [F (1,97) = 27.3, *p* < 0.00001] and *Fgfr1* [F (1,45) = 6.14, *p* < 0.05]. There was no sex difference in *Fgf21* expression [F (1,47) = 3.75, *p* = 0.059].

In a separate analysis in male mice, oral RK had no significant effect on gene expression of *Pparα*, *PGC1α*, *Fgf21* and *Fgfr1* in each diet group (Appendix A). HFD increased *Pparα* expression in mice that received RK; KD mice had reduced *Fgf21* expression compared to HFD or CD mice; KD or CD mice had higher *Fgfr1* expression. *PGC1α* expression was not affected by RK or diet.

In female mice, the gene expressions of *Ppara* and *PGC1α* were promoted in KD mice compared to HFD mice, and in RK-dosed mice, the *Fgf21* expression was increased in CD mice compared to HFD or KD mice. There were no effects of RK or diet on *Fgfr1* expression (Appendix A).

There were sex effects for circulating apelin [F (1,87) = 106.4, *p* < 0.0001], insulin [F (1,93) = 28.9, *p* < 0.0001], and leptin [F (1,90) = 69.8, *p* < 0.0001]. Male mice had higher levels of apelin, insulin, and leptin than female mice (*p* < 0.05).

In male mice, oral RK had no effect on plasma apelin and leptin levels in each diet group (Figure 3A,E). Insulin levels were decreased by RK in HFD mice (*p* < 0.001; Figure 3C). Diets had a major impact on all these plasma protein levels. CD mice had the highest (*p* < 0.01), while HFD mice had the lowest plasma apelin levels (*p* < 0.01; Figure 3A). In vehicle-dosed mice, HFD mice had higher insulin levels compared to KD or CD mice (*p* < 0.0001). However, in RK-dosed mice, there is no diet effect on insulin levels (Figure 3C). For plasma leptin levels, HFD mice had increased leptin compared to KD or CD mice (*p* < 0.0001). KD mice had slightly higher leptin levels than CD mice (*p* < 0.05). However, this was only found in vehicle-dosed mice but not in RK-dosed mice (Figure 3E).

In female mice, RK had no effect on apelin levels in each diet group. All CD mice had higher apelin levels than HFD (*p* < 0.01) or KD (*p* < 0.01) mice. There were no differences in apelin levels between HFD and KD mice (Figure 3B). Insulin levels were reduced by RK in CD mice (*p* < 0.01). Vehicle-dosed CD mice had higher insulin levels than vehicle-dosed HFD or KD mice (*p* < 0.01; Figure 3D). Leptin levels in female mice were not affected by RK or diets (Figure 3F).

### 3.4. Effects of Daily Raspberry Ketone Dosing and Diets Feeding on Overall Gut Microbiota Diversity and Structure

To explore whether RK dosing and diets feeding modify gut microbiota diversity and structure, we analyzed gut microbiota from RK or Veh-dosed KD and CD mice (n = 5/group) by sequencing hypervariable region V4 of 16S rRNA gene. A total of 474 amplicon sequence variants (ASVs) were retained for analysis. We used the Shannon index as the representative metric for alpha diversity (Figure 4A). The gut microbiota of KD_Veh male mice had significantly higher alpha diversity than KD_Veh female mice and CD-fed mice (CD_Veh and CD_RK) (*p* < 0.05; Figure 4A). However, RK dosing had no significant effect on alpha diversity compared to vehicle in each diet group in male or female mice.

Principal coordinates analysis plot of beta diversity based on the Bray Curtis distance was plotted to visualize differences in overall gut microbiota structure (Figure 4B). Permutational multivariate analysis of variance (PERMANOVA) showed that beta diversity of gut microbiota was impacted by sex [F (1,32) =14.8, *p* < 0.001], diet [F (1,32) = 12.9, *p* < 0.005], dosing*sex [F (1,32) = 2.83, *p* < 0.002] and diet*sex [F (1,32) = 5.62, *p* < 0.001]. In male mice, there is an obvious separation between KD and CD-fed mice (*p* < 0.009). The gut microbiota of RK-dosed mice also had a significant difference from the vehicle in KD-fed (*p* = 0.018) but not in CD-fed mice. In female mice, KD and CD-fed mice had significantly different gut microbiota (*p* < 0.05). However, RK dosing did not impact gut microbiota in female mice.

## 4. Discussion

This study sought to examine the additive effects of RK oral supplementation with KD-induced body weight loss and related metabolic improvements. To examine this, male and female C57Bl/6J mice were fed an HFD for 9 weeks and switched to a KD (80% fat) or CD (10%) or were maintained on an HFD. At the time of the diet switch, mice were orally dosed with RK (200 mg/kg) or vehicle per day for 4 weeks. Our major findings were sex-specific. In males on KD, daily dosing with RK reduced body weight and body fat compared with vehicles. In females on KD, there were effects on glucose homeostasis with RK compared with vehicles. Indeed, there was a lower area under the curve following the oral glucose tolerance test with RK dosing compared with the vehicle. Hepatic gene expression and metabolic hormonal profile did not uncover any differences between RK and KD. There were, however, differences in the gut microbiota with KD and with RK dosing. Overall, oral RK 200 mg/kg resulted in mild to moderate improvements in KD-induced weight loss outcomes.

In order to promote diet-induced ketosis, KDs are high in fat and low in carbohydrate content and have been demonstrated to reduce body weight and improve metabolic outcomes in rodents and humans [27,28,29]. In this study, KD exposure began after 9 weeks on HFD. Overall, there was not a body weight difference in the mice switched to the KD and those mice that continued on the HFD. One explanation for this was that there was a similar caloric intake from the HFD and KD diets. Blood ketone levels, however, were elevated by KD in both male and female mice. Notably, female mice had higher ketone levels, and there were sex-specific effects on hepatic gene expression of *Pparα*, *PGC1α* and *Fgfr1*. However, the expression pattern of these genes in males or females does not seem to be an explicit explanation for increased ketone levels by KD. In addition, KD feeding led to impaired glucose tolerance. This was also observed in several other KD-related studies [20,30]. The mechanism of KD-induced glucose intolerance is not clear. KD feeding induced decreased GLUT4 expression and reduced muscle glucose uptake [31], reduced α-and β-cell mass and insufficient insulin [19], and changes in gut microbiota and metabolites [30] may contribute to glucose intolerance and insulin resistance. In this study, we analyzed several plasma hormones that are involved in glucose and lipid metabolism regulation. There is a sex difference in the circulation of apelin, insulin and leptin levels: overweight male mice had higher levels than overweight female mice. This may be because overweight male mice had larger body weight/size (33.3 ± 0.7 g) than female mice (25.3 ± 0.3 g) in this study or different sensitivities to these hormones compared with female mice. Circulating leptin levels were relatively proportional to changes in body weight and fat in this study. Insulin level in KD mice is lower than in HFD mice in males or CD mice in females. In addition, apelin, which could lower glucose and stimulate muscle glucose uptake [32], is lower in KD mice. The alterations of glucose metabolism-related hormones by KD feeding may contribute to the hyperglycemia and impaired glucose tolerance observed in KD mice.

As a dietary supplement, raspberry ketone has been found to prevent HFD-induced body weight gain and reduce fat mass in our previous studies and others [5,6,8]. We also found that RK reduced hyperglycemia during the carbohydrates tolerance test, which may attribute to its α-glucosidase inhibitor properties [5]. In this study, RK was orally dosed daily to examine weight loss and glucose-lowering effects in mice with weight gain caused by HFD feeding. Previously we demonstrated the action of RK on fat loss may be associated with changes in adipose apelin gene expression, as reported previously [5]. Unexpectedly, the addition of RK-induced loss of lean mass in KD mice may also contribute to the weight loss observed in KD_RK mice. RK’s effect on loss of weight, fat and lean mass was not observed in HFD or CD-fed mice, suggesting a diet or energy source-dependent effect. In addition, RK daily dosing modeled gut microbiota structure and changed the beta diversity of gut microbiota. These may all contribute to RK’s body weight gain prevention effect, as we observed in a previous study [5,7].

*Strengths and limitations.* In this study, we aimed to investigate the effects of the addition of RK to KD feeding on body weight, body composition, and glucose tolerance. One strength is that we used HFD-induced overweight/obese male and female mice to investigate the impact of diet and supplementation on weight loss and glucose metabolism. Despite our findings on body weight and glycemic levels, the diet and treatment period were only 4 weeks in duration. Previous findings from our lab have indicated RK is more effective at *preventing* HFD-induced weight gain than reducing weight gain [5]. Although oral dosing with RK reduced the percentage of body fat, there was also a reduction in the percentage of lean mass with RK (see Appendix A) that could have contributed to RK’s effects on body weight. However, this effect on lean mass could be a transitory effect on body composition. This notion can only be supported with additional long-term studies to fully investigate the effects of RK with KD on body composition. Since this is the first study of RK on KD outcomes, we did not conduct a more targeted approach to investigate the mechanisms of action of RK on body composition and gut microbiota phylotypes. Another limitation of the current study is that we only use a single concentration of 200 mg/kg of RK. The choice for this dose was based on our previous findings suggesting that 200 mg/kg prevents HFD-induced weight gain [5,6,7]. Nonetheless, a full dose-dependent effect of RK on KD should be investigated. The dose-dependent effects of RK on KD are a critical factor in elucidating any potential additive effect on KD outcomes. Previous work from our lab and others has suggested that oral doses of RK > 500 mg/kg in mice can produce acute toxicity [33,34,35]. Further work in our lab is being performed to determine whether prolonged exposure to a ketogenic diet and exogenous ketones promotes a higher toxicity threshold for RK.

## 5. Conclusions

The additive effects of oral supplementation of RK on KD weight and fat loss effects in males but improves glycemic outcomes in female mice. The independent sex-specific effects of RK to prevent fat accumulation and shifts in gut microbiota structure modeling may strengthen RK’s beneficial impacts on weight loss with KD. Further work is needed to determine the dose-dependent effects of RK and to elucidate the toxicity endpoints.

## Data Availability

The raw data supporting the conclusions of this article will be made available by the authors without undue reservation.

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
