# Peer review of "Sex-Specific Effects on Total Body Fat Gain with 4-Week Daily Dosing of Raspberry Ketone [4-(4-Hydroxyphenyl)-2-butanone] and Ketogenic Diet in Mice"

_nutrients, 2023, doi:10.3390/nu15071630_

Round 1

Reviewer 1 Report

The manuscript describes a study that aimed to investigate the potential additive effects of oral raspberry ketone (RK) supplementation on weight loss outcomes in high-fat diet (HFD)-fed mice, particularly in the context of a ketogenic diet (KD). The study found that, in male mice, RK supplementation reduced body fat and elevated endogenous ketone body levels, but these effects were not influenced by diet. Furthermore, while switching to KD resulted in greater weight loss and fat mass reduction compared to HFD, RK supplementation did not further improve KD's weight loss effect or high-fat diets feeding-induced glucose intolerance.

Overall, the manuscript appears to be well-written and very complex; However, there are some limitations and potential areas for improvement that could be addressed in the manuscript.

Another potential area for improvement is the clarity of the manuscript's organization. The introduction and methods sections could be more clearly separated to better describe the rationale for the study and the methods used to carry it out. Additionally, the results section could be more clearly structured to highlight the main findings and their implications.

The study did not explore the effects of varying doses or duration of RK supplementation, as only one dose was used. As a result, the conclusion drawn from the study may be impacted by this limitation. It is possible that higher doses of RK could produce a more substantial increase in ketone levels and lead to greater effects on weight loss and glucose tolerance. 

The presentation of the figures is rather confusing. Also, the authors should use asterisks ("stars") or the actual p value to indicate statistical significance in the figures. 

Did the authors check the ghrelin levels?

It is surprising that the RK supplementation resulted in only a modest increase in ketone levels and may have impacted the study's results; i.e this increase was not sufficient to produce any additional weight loss or improve glucose tolerance?

The authors should have highlighted the sex differences in their study, as these were mentioned in the title.

It appears that the manuscript does not include a section on the limitations of the study, which is an important aspect of any research publication to address any potential biases, weaknesses, or limitations that may have influenced the findings.

Please provide clinical takeaways or implications that can be drawn from the results of this study

Author Response

Reviewer 1

The manuscript describes a study that aimed to investigate the potential additive effects of oral raspberry ketone (RK) supplementation on weight loss outcomes in high-fat diet (HFD)-fed mice, particularly in the context of a ketogenic diet (KD). The study found that, in male mice, RK supplementation reduced body fat and elevated endogenous ketone body levels, but these effects were not influenced by diet. Furthermore, while switching to KD resulted in greater weight loss and fat mass reduction compared to HFD, RK supplementation did not further improve KD's weight loss effect or high-fat diets feeding-induced glucose intolerance.

Overall, the manuscript appears to be well-written and very complex; However, there are some limitations and potential areas for improvement that could be addressed in the manuscript.

Authors Response: Thanks for your comments and suggestions. We made major changes to the Results presentation and the Discussion.

Another potential area for improvement is the clarity of the manuscript's organization. The introduction and methods sections could be more clearly separated to better describe the rationale for the study and the methods used to carry it out. Additionally, the results section could be more clearly structured to highlight the main findings and their implications.

Authors Response: Thanks. We added more background information in Introduction, in Materials and Methods section a Schematic Figure for experimental design; redid statistical analysis and graphing the Figures for results.  This should provide more clarity.

The study did not explore the effects of varying doses or duration of RK supplementation, as only one dose was used. As a result, the conclusion drawn from the study may be impacted by this limitation. It is possible that higher doses of RK could produce a more substantial increase in ketone levels and lead to greater effects on weight loss and glucose tolerance.

Authors Response: Yes, we are limited by using a single dose of raspberry ketone. Based our previous studies in mice, the 200 mg/kg was chosen for this study. This is nor outlined in the Introduction. We have stated the limitation of using a single dose and potential limitation of using a higher dose in the Discission section.

The presentation of the figures is rather confusing. Also, the authors should use asterisks ("stars") or the actual p value to indicate statistical significance in the figures. 

of the figures is rather confusing.

Authors Response: Yes, we have changes to the figures and used asterisks to indicate level of significance.

Did the authors check the ghrelin levels?

Authors Response: We did not check the ghrelin levels in this study. In our previous study (Kshatriya D et al, 2019), in high fat diet fed male mice, ghrelin levels were reduced after 4-week RK daily dosing. In that paper, however, the reduction of ghrelin was not related to RK treatment. Since the overall caloric intake were similar in the HFD and KD in the present study, we did not measure ghrelin levels.  

It is surprising that the RK supplementation resulted in only a modest increase in ketone levels and may have impacted the study's results; i.e this increase was not sufficient to produce any additional weight loss or improve glucose tolerance?

Authors response: Yes, we have noted this in the Discussion.

The authors should have highlighted the sex differences in their study, as these were mentioned in the title.

Authors Response: Thanks. In the revised manuscript we have highlighted the sex differences in Results and Discussion.

It appears that the manuscript does not include a section on the limitations of the study, which is an important aspect of any research publication to address any potential biases, weaknesses, or limitations that may have influenced the findings.

Authors Response: Thanks. We have added sections for limitations and strengths in Discussion.

Please provide clinical takeaways or implications that can be drawn from the results of this study

Authors Response: We have added the following to the Conclusion section (lines 402-407)

“The additive effects of oral supplementation of RK on KD weight and fat loss effects in males but improves glycemic outcomes in female mice. The independent sex-specific effects of RK to prevent fat accumulation and shifts in gut microbiota structure modeling may strengthen RK’s beneficial impacts on weight loss with KD. Further work is needed to determine the dose dependent effects of RK and to elucidate the toxicity endpoints.”  

Reviewer 2 Report

This study aims to demonstrate the added value that the addition of an oral ketone (RK) can bring when using a KD diet in animals that have developed obesity. This study is also original in the sense that it tests both males and females.

 Introduction:

-          More details on the RK are needed, especially where are they found? In what type of foods? Are they only produced by industry?

-          Line 30-33 “This very low carbohydrate and very high fat diet, originally designed for neurological diseases, has demonstrated a weight loss effect and has been used for dietary intervention in weight loss studies for obesity.” Do you have some references?

-          The introduction is short. You should add more context, especially about ketogenic diet, and its beneficial effects on weight loss. What about the molecular effects of KD and RK?

-          In your hypothesis line 44-46 “We examined in current study whether the additive of RK to ketogenic dietary intervention would augment the beneficial effects or ameliorate the adverse effects of KD diet on bodyweight loss and glucose homeostasis”, you talk about the beneficial effect of KD, and adverse effects, but they don’t figure in you introduction. In my opinion, you should add a paragraph, where you explain, with scientific studies, the beneficial effects of KD in weight loss, and its limits.

-          Did you expect different results between male and females?

-          Ligne 45: “of of”.

Materials and methods:

-          A schematic figure of the experimental protocol design, clearly showing the different groups and the different diets, should appears in the introduction. The explanation of the different groups with only a text is hard to follow.

Results:

-          “Different letters indicate statistical difference”. It is not clear. You should clearly explain your captions. For each of your letters, what do they correspond to?

You should add more precisions on it, because it is really hard to understand your statistical comparisons.

-          What about the results concerning the food intake and calories count between groups? A figure could be welcome for these important results. Theses results have to be discussed.

-          I don't understand how the males in the KD-RK group can lose double the weight every day and there is no difference at the end of the 4 weeks? Can you add the curves of weight evolution during the protocol?

-          why did the animals in the HFD groups lose weight? especially in the last 10 days. One would have imagined that these animals continued to gain weight in these groups.

-          Concerning table 1, it would be clearer to highlight the main results on small figures rather than a large table complicated to interpret

-          table 2 could also be transformed into a figure to be more clearly visualized, or perhaps figure 2 seems to be enough?

-          Same remarks for table 3 and 4.

-          The description of your results is tedious to read, because of the many groups and the many comparisons. You should correct them, describing them group by group, and indicating which group you are comparing it to. It would be clearer.

-          It would have been interesting to look in qPCR at the expression of the other targets of PPARa which are involved in metabolism, such as the beta-oxidation or lipogenesis genes.

-          How can you explain the difference of insulin levels between males and females?

Discussion:

-          The results concerning the difference of diet effects between males and females are very interesting and have to be more discuss and need to be better highlighted.

-          How can you explain the loss of lean mass in your model? The energy is not enough in the CD?

-          In your discussion, you do not mention your results obtained at the molecular level, about FGF21. Still, they are surprising because other publications have shown an increase in FGF21 expression during a ketogenic diet. How do you explain that? (Cf : https://doi.org/10.1016/j.cmet.2007.05.002 ; https://doi.org/10.1210/en.2009-0532 ; https://doi.org/10.1016/j.nut.2021.111230)

-          Concerning the conclusion, normally a dose response to determine the potential toxic effects of RK should have been carried out before this type of study because otherwise it is difficult to conclude on the effects of the molecule...

-          The conclusion could also be more focused on the results concerning the differences between males and females. Indeed, RK seems to have greater effects in males than in females.

Author Response

Reviewer 2

This study aims to demonstrate the added value that the addition of an oral ketone (RK) can bring when using a KD diet in animals that have developed obesity. This study is also original in the sense that it tests both males and females.

Introduction:

More details on the RK are needed, especially where are they found? In what type of foods? Are they only produced by industry?

Authors Response: Thanks. We have added more details of RK in the Introduction section.

Line 30-33 “This very low carbohydrate and very high fat diet, originally designed for neurological diseases, has demonstrated a weight loss effect and has been used for dietary intervention in weight loss studies for obesity.” Do you have some references?

Authors Response: Thanks. We modified the sentence and provided references.

The introduction is short. You should add more context, especially about ketogenic diet, and its beneficial effects on weight loss. What about the molecular effects of KD and RK?

Authors Response: Thanks. We added more background information for ketogenic diet and raspberry ketone. Please tracked changes in Introduction.  

In your hypothesis line 44-46 “We examined in current study whether the additive of RK to ketogenic dietary intervention would augment the beneficial effects or ameliorate the adverse effects of KD diet on bodyweight loss and glucose homeostasis”, you talk about the beneficial effect of KD, and adverse effects, but they don’t figure in you introduction. In my opinion, you should add a paragraph, where you explain, with scientific studies, the beneficial effects of KD in weight loss, and its limits.

Authors Response: Thanks. We added in both beneficial effects and limits of KD in weight loss and glucose homeostasis regulation in the Introduction.

Did you expect different results between male and females? 

Authors Response: Yes, we examined sex as a variable in this study.

Ligne 45: “of of”.

Authors Response: Thanks. We have corrected this error.

Materials and methods:

A schematic figure of the experimental protocol design, clearly showing the different groups and the different diets, should appears in the introduction. The explanation of the different groups with only a text is hard to follow.

Authors Response: Thanks. We added a schematic diagram for experimental design. Please see section 2.1 of Materials and Methods.

Results:

 “Different letters indicate statistical difference”. It is not clear. You should clearly explain your captions. For each of your letters, what do they correspond to?

You should add more precisions on it, because it is really hard to understand your statistical comparisons.

Authors Response: Thanks. We regraphed our figures and added more precisions for statistical comparisons.  

What about the results concerning the food intake and calories count between groups? A figure could be welcome for these important results. Theses results have to be discussed.

Authors Response: Thanks. We have added figures to show these results. Please see Figure 1 G and 1H.

I don't understand how the males in the KD-RK group can lose double the weight every day and there is no difference at the end of the 4 weeks? Can you add the curves of weight evolution during the protocol?

Authors Response: Thanks for your comments. We analyzed bodyweight change at day 30 rather than raw bodyweight using diet and treatment as independent variables in two-way ANOVA analysis. We found a weight loss in RK dosed male KD mice but not in HFD or CD mice. This is graphed and represented in Figure 1C and 1D (p<0.05). We confirmed these results by analyzing weight changes of mice (at Day 22) prior fecal collection, ketone measures, oral glucose tolerance test and body composition measures which may introduce stresses and affect bodyweight. The results showed that RK dosing at day 22 reduced bodyweight in KD mice (p<0.01; data not shown).

why did the animals in the HFD groups lose weight? especially in the last 10 days. One would have imagined that these animals continued to gain weight in these groups.

Authors Response: All mice in this study were fed HFD (45% fat) for 9 weeks without oral dosing. All mice were then switched to assigned diets and received daily oral gavage of RK or vehicle for 4 weeks. The caloric intake were the same between HFD and KD. Mice were also subjected to fecal collection, ketone measures and oral glucose tolerance test at day 23; body composition measures at day 28. All these experimental manipulations are disruptive to the homeostatic controls of body weight for mice. We have noted the similar body weights in the Discussion section. (lines 342-345) 

“In this study, KD exposure began after 9 weeks on HFD. Overall, there was not a body weight difference in the mice switched to the KD and those mice that continued on the HFD. One explanation for this was that there was a similar caloric intake from the HFD and KD diets.”

 Concerning table 1, it would be clearer to highlight the main results on small figures rather than a large table complicated to interpret

Authors Response: Thanks. We reanalyzed all data and highlighted the main results in Figures.

Table 2 could also be transformed into a figure to be more clearly visualized, or perhaps figure 2 seems to be enough?

Authors Response: Thanks. We have transformed Table to into Figures. Please see Figure 2 and others.

Same remarks for table 3 and 4.

Authors Response: Thanks. We have transformed Table to into Figures.

The description of your results is tedious to read, because of the many groups and the many comparisons. You should correct them, describing them group by group, and indicating which group you are comparing it to. It would be clearer.

Authors Response: Thank you. This was an excellent observation and we have reorganized our figures and contexts for comparisons.  

It would have been interesting to look in qPCR at the expression of the other targets of PPARa which are involved in metabolism, such as the beta-oxidation or lipogenesis genes.

Authors Response: Yes, for this initial study we were limited in our gene targets. We agree a more comprehensive analysis is needed for future studies.  

How can you explain the difference of insulin levels between males and females?

Authors Response: We noted there a difference in the hormones we measured based on sex in the Discussion (lines 353-359). 

Discussion:

The results concerning the difference of diet effects between males and females are very interesting and have to be more discuss and need to be better highlighted.

Authors Response: Thanks for your comments. We have added more discussions regarding the sex different effects of diets.

How can you explain the loss of lean mass in your model? The energy is not enough in the CD?

Authors Response: Thanks. In this study, we used three diets with different fat and carbohydrates composition: HFD (45% fat and 35%carbohydrate), KD (80% fat and 0.1% carbohydrate) and CD (10%fat and 70%carbohydrate). All diets have 20% energy from protein. As shown in our results (Figure 1G and H), CD mice have less accumulated diet intake than HFD or KD mice in both male and female mice. Accordingly, mice may have less energy intake from protein which is also important for lean mass maintenance. Importantly, these animals were fed a HFD for 9 weeks prior to beginning KD and CD.

In your discussion, you do not mention your results obtained at the molecular level, about FGF21. Still, they are surprising because other publications have shown an increase in FGF21 expression during a ketogenic diet. How do you explain that? (Cf : https://doi.org/10.1016/j.cmet.2007.05.002 ; https://doi.org/10.1210/en.2009-0532 ; https://doi.org/10.1016/j.nut.2021.111230)

Authors Response: Yes, this is a good point. At this point, this might be outside the scope of our initial study. The role plasma FGF21 levels is something we will definitely consider for future experiments. 

-          Concerning the conclusion, normally a dose response to determine the potential toxic effects of RK should have been carried out before this type of study because otherwise it is difficult to conclude on the effects of the molecule...

Authors Response: Thank you. This a clear limitation of the study. We have acknowledged in the Discussion section. (Lines 392-399)

The conclusion could also be more focused on the results concerning the differences between males and females. Indeed, RK seems to have greater effects in males than in females.

Authors Response: Yes, we have attempted to broaden the discussion of the male and female differences.

Round 2

Reviewer 1 Report

The authors have addressed all the concerns raised in my previous review, and the revisions have strengthened the clarity and quality of their work. I have no further comments to add.